# Face-to-face panel meetings versus remote evaluation of fellowship applications: simulation study at the Swiss National Science Foundation

Marco Bieri ![ORCID],[1] Katharina Roser ![ORCID],[1,2] Rachel Heyard ![ORCID],[3] Matthias Egger ![ORCID] [4,5,6]

MB and KR contributed equally.

¹Careers Division, Swiss National Science Foundation, Bern, Switzerland
²Department of Health Sciences and Medicine, University of Lucerne, Lucerne, Switzerland
³Data Team, Swiss National Science Foundation, Bern, Switzerland
⁴Institute of Social & Preventive Medicine, University of Bern, Bern, Switzerland
⁵Population Health Sciences, Bristol Medical School, University of Bristol, Bristol, UK
⁶Research Council, Swiss National Science Foundation, Bern, Switzerland

**Correspondence to**
Dr Marco Bieri;
marco.bieri@snf.ch

## ABSTRACT

**Objectives** To trial a simplified, time and cost-saving method for remote evaluation of fellowship applications and compare this with existing panel review processes by analysing concordance between funding decisions, and the use of a lottery-based decision method for proposals of similar quality.

**Design** The study involved 134 junior fellowship proposals for postdoctoral research ('Postdoc. Mobility'). The official method used two panel reviewers who independently scored the application, followed by triage and discussion of selected applications in a panel. Very competitive/uncompetitive proposals were directly funded/rejected without discussion. The simplified procedure used the scores of the two panel members, with or without the score of an additional, third expert. Both methods could further use a lottery to decide on applications of similar quality close to the funding threshold. The same funding rate was applied, and the agreement between the two methods analysed.

**Setting** Swiss National Science Foundation (SNSF).

**Participants** Postdoc.Mobility panel reviewers and additional expert reviewers.

**Primary outcome measure** Per cent agreement between the simplified and official evaluation method with 95% CIs.

**Results** The simplified procedure based on three reviews agreed in 80.6% (95% CI: 73.9% to 87.3%) of applicants with the official funding outcome. The agreement was 86.6% (95% CI: 80.6% to 91.8%) when using the two reviews of the panel members. The agreement between the two methods was lower for the group of applications discussed in the panel (64.2% and 73.1%, respectively), and higher for directly funded/rejected applications (range: 96.7%–100%). The lottery was used in 8 (6.0%) of 134 applications (official method), 19 (14.2%) applications (simplified, three reviewers) and 23 (17.2%) applications (simplified, two reviewers). With the simplified procedure, evaluation costs could have been halved and 31 hours of meeting time saved for the two 2019 calls.

**Conclusion** Agreement between the two methods was high. The simplified procedure could represent a viable evaluation method for the Postdoc.Mobility early career instrument at the SNSF.

## Strengths and limitations of this study

► The study examined the outcome between a simplified and the official evaluation procedure for junior fellowship applications for different research disciplines.

► The study discussed the agreement between the two evaluation methods in the context of the general uncertainty around peer review and estimated the costs and time that could have been saved with the simplified evaluation procedure.

► It is the first study to provide insight into lottery-based decisions in the context of the evaluation of junior fellowship applications.

► The study lacks statistical power because the number of applications was relatively small.

► The study addressed the specific context and evaluation of the Swiss National Science Foundation Postdoc.Mobility funding scheme, results may thus not be generalisable to other funding programmes.

## INTRODUCTION

Peer review of grant proposals is costly and time-consuming. The burden on the scientific system is increasing, affecting funders, reviewers and applicants.[1 2] In response, researchers have studied the review process and examined simplifications. For example, Snell[3] studied the number of reviewers and consistency of decisions and found that five evaluators represented an optimal tradeoff. Graves *et al*[4] assessed the reliability of decisions made by evaluation panels of different sizes. They concluded that reliability was greatest with about 10 panel members. Herbert *et al*[5] compared smaller panels and shorter research proposals with the standard review procedure. The agreement was about 75% between simplified and standard procedures. As an alternative to face-to-face (FTF) panels, the use of virtual, online meetings has also been examined. Bohannon[6] reported that at the National Science Foundation and National Institutes of Health (NIH),

virtual meetings could reduce costs by one-third. Gallo et al[7] compared teleconferencing with FTF meetings and found only few differences in the scoring of the applications. Later studies also found that virtual and FTF panels produce comparable outcomes.[8–10]

With virtual formats, panel members still need to attend time-consuming meetings. Using the reviewers' written assessments without FTF or virtual panel discussions would simplify the process further. Fogelholm et al[11] reported that results were similar when using panel consensus or the mean of reviewer scores. Obrecht et al[12] noted that panel review changed the funding outcome of only 11% of applications. Similarly, Carpenter et al[8] found that the impact of discussions was small, affecting the funding outcome of about 10% of applications. Pina et al[13] studied Marie Curie Actions applications and concluded that ranking applications based on reviewer scores might work for some but not all disciplines. In the Humanities, Social and Economic Sciences, an exchange between reviewers may be particularly relevant. The triaging of applications has also been examined: after an initial screening, non-competitive and very competitive proposals are either directly rejected or funded. Vener et al[14] validated the triage model of the NIH and found that the likelihood of erroneously discarding a competitive proposal was very small. Bornmann et al's[15] findings on a multistage fellowship selection process also supported the use of a triage.

Mandated by the government, the Swiss National Science Foundation (SNSF) is Switzerland's foremost funding agency, supporting scientific research in all disciplines. Following changes to the career-funding portfolio, the SNSF will experience a significant increase of applications for the junior 'Postdoc.Mobility' fellowship scheme, which offers postdoctoral researchers a stay at a research institution abroad for up to 24 months. The scheme enables junior postdocs to deepen their scientific knowledge and increase their scientific independence during a research stay abroad. The aim of this work was to compare the evaluation of applications by expert review, triage, and discussion in an evaluation panel with expert reviews only.

## METHODS

### Sample

We included applications submitted for the August 2019 Postdoc.Mobility fellowship call. We also included applications by Postdoc.Mobility fellows for a return grant to facilitate their return to Switzerland. Both fellowship and return grants were evaluated according to the same criteria by one of five panels: Humanities, Social Sciences, STEM (Science, Technology, Engineering and Mathematics), Biology or Medicine.

### Study design

We compared funding outcomes based on the official, legally binding evaluation with a simulated, hypothetical evaluation. The official evaluation was based on the triage of applications based on expert reviews, followed by a discussion of the meritorious applications in an FTF panel: the triage-panel meeting (TPM) format (figure 1). In a first step, each proposal was independently reviewed and scored by two panel members. For the assessment, the evaluation criteria defined in the Postdoc.Mobility regulations[16] were applied. The criteria address different aspects of the applicant, the proposed research project and the designated research location. Panel members used a 6-point scale: outstanding=6 points, excellent=5 points, very good=4 points, good=3 points, mediocre=2 points and poor=1 point. Applications were then allocated to three groups based on the ranking of the mean scores given to each proposal: Fund without further discussion (F in figure 1), Discuss in panel meeting (D) and Reject (R). Panel members could request that applications in the F or R group are reallocated to D and discussed. In a second step, the D proposals were discussed in the FTF panel meeting, ranked and funded or rejected. Random selection (RS in figure 1) could be used to fund or reject proposals of similar quality close to the funding threshold if the panel could not reach a decision. Funding decisions were based on the standard two-stage method, which included FTF panel meetings (TPM).

The simulated alternative procedure consisted only of the first step, that is, was entirely based on the ranking of proposals based on the expert reviews: the expert review-based (ERB) evaluation. In addition to the two panel members, a third expert reviewer who was not a member of the panel assessed the proposal. The same 6-point scale was used. The proposals were then allocated to one of three groups based on the mean scores (F, RS and R in figure 1). Random selection was used whenever the funding line went through a group of two or more applications with identical scores. The funding rate of the TPM was applied to the simulated ERB method.

### Data analysis

To determine the agreement between the two evaluation methods, we used 2×2 contingency tables. We calculated the simple agreement with 95% CIs, which were generated using a bootstrap algorithm.[5] We also examined the agreement between the TPM and the ERB approach using only the assessments from the two panel members, thus excluding the assessment from the third reviewer. We calculated discipline-specific and gender-specific levels of agreement. We used $\chi^2$ tests for categorical data to test whether the agreement differed between these mutually exclusive groups.

### Costs

We determined the costs related to the evaluation. The costs comprised expenses related to the scientific assessment of the individual applications and the panel meetings. The SNSF compensates panel reviewers with US$275 per scientific assessment. Panel reviewers further receive a meeting allowance of up to US$550 depending on the

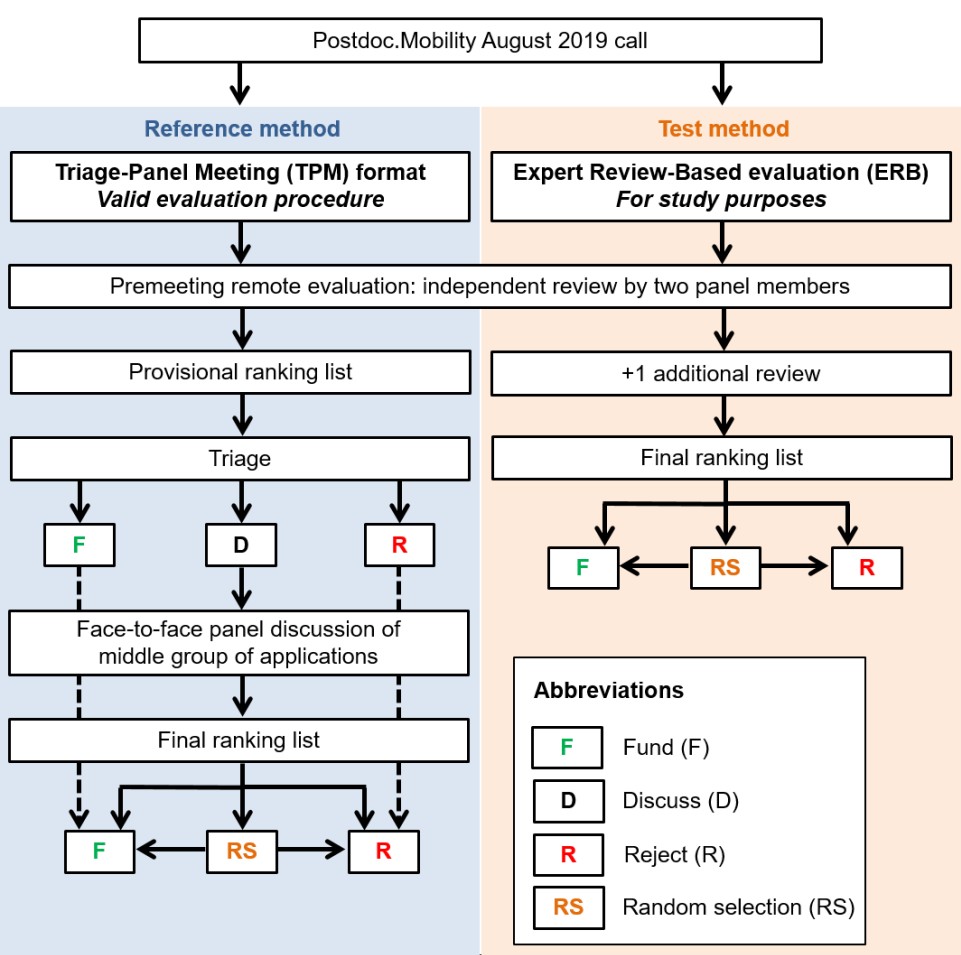

**Figure 1** Design of the study comparing the ERB evaluation with the TPM format. The ERB and the TPM were dependent in terms of the two assigned panel reviewers per application. The third reviewers were only added for the ERB, their assessments were not considered for the TPM and therefore the official funding outcome. ERB, expert review based; TPM, triage-panel meeting.

duration of the meeting. Further, the SNSF reimburses travel expenses and accommodation costs. The five panels included 96 members and met twice in 2019.

### Patient and public involvement
This analysis was based on expert review reports submitted to the SNSF. No patients were involved in developing the research question, outcome measures and overall design of the study.

## RESULTS
### Study sample and success rates
The sample consisted of 134 applications, including 124 fellowship applications and 10 requests for a return grant. The mean age of applicants was 32.7 years (SD 3.2 years) in men and 33.5 years (SD 2.8 years) in women. Each reviewer received a mean of 2.5 (SD 1.4) applications to evaluate.

Table 1 shows the summary statistics of applications and success rates across disciplines, genders and the three evaluation methods: the legally binding TPM format and the simulated ERB evaluations with three or two reviewers.

Most applications came from biology, followed by the STEM disciplines and the Social Sciences. Almost two-thirds of applications came from men. With TPM, success rates were slightly higher in women (60.4%) than in men (50.0%). This was driven by the middle group of applications that were discussed in the panels, where the success rate of women overall was 66.7% (24 of 67 applicants were women in this group). Success rates were similar across disciplines, ranging from 56.2% in the Humanities to 52.2% in the Social Sciences. By design, overall success rates were the same with the ERB evaluations; however, the difference between genders was smaller with ERB than with TPM (table 1).

### Agreement between evaluation by ERB or TPM
Comparing the ERB evaluation based on three reviewers with the standard TPM format, the agreement overall was 80.6% (95% CI: 73.9% to 87.3%). The agreement was highest in the Medicine panel (90.0%; 95% CI: 75% to 100%) and lowest in the Social Sciences panel (73.9%; 95% CI: 56.5% to 91.3%). However, the statistical evidence for differences in agreement between panels was

**Table 1** Success rates by gender of applicants, by discipline and type of evaluation

| Discipline | All applicants | | Women | | Men | |
|---|---|---|---|---|---|---|
| | N | N funded (%) | N | N funded (%) | N | N funded (%) |
| **TPM** | | | | | | |
| All disciplines | 134 | 72 (53.7) | 48 | 29 (60.4) | 86 | 43 (50.0) |
| Humanities | 16 | 9 (56.2) | 9 | 4 (44.4) | 7 | 5 (71.4) |
| Social Sciences | 23 | 12 (52.2) | 10 | 7 (70.0) | 13 | 5 (38.5) |
| STEM | 35 | 19 (54.3) | 10 | 6 (60.0) | 25 | 13 (52.0) |
| Biology | 40 | 21 (52.5) | 14 | 8 (57.1) | 26 | 13 (50.0) |
| Medicine | 20 | 11 (55.0) | 5 | 4 (80.0) | 15 | 7 (46.7) |
| **ERB (three reviewers)*** | | | | | | |
| All disciplines | 134 | 72 (53.7) | 48 | 27 (56.3) | 86 | 45 (52.3) |
| Humanities | 16 | 9 (56.3) | 9 | 5 (55.6) | 7 | 4 (57.1) |
| Social Sciences | 23 | 12 (52.2) | 10 | 6 (60.0) | 13 | 6 (46.2) |
| STEM | 35 | 19 (54.3) | 10 | 4 (40.0) | 25 | 15 (60.0) |
| Biology | 40 | 21 (52.5) | 14 | 8 (57.1) | 26 | 13 (50.0) |
| Medicine | 20 | 11 (55.0) | 5 | 4 (80.0) | 15 | 7 (46.7) |
| **ERB (two reviewers)†** | | | | | | |
| All disciplines | 134 | 72 (53.7) | 48 | 25 (52.1) | 86 | 47 (54.7) |
| Humanities | 16 | 9 (56.3) | 9 | 5 (55.6) | 7 | 4 (57.1) |
| Social Sciences | 23 | 12 (52.2) | 10 | 6 (60.0) | 13 | 6 (46.2) |
| STEM | 35 | 19 (54.3) | 10 | 4 (40.0) | 25 | 15 (60.0) |
| Biology | 40 | 21 (52.5) | 14 | 7 (50.0) | 26 | 14 (53.8) |
| Medicine | 20 | 11 (55.0) | 5 | 3 (60.0) | 15 | 8 (53.3) |

*Two of the three expert reviewers were also members of the evaluation panel.
†Both expert reviewers were also members of the evaluation panel.
ERB, expert review based; N, number of applications; STEM, Science, Technology, Engineering, Mathematics; TPM, triage-panel meeting.

weak (p=0.58, table 2). As expected, the agreement was higher when comparing the ERB evaluation based on the two panel members with TPM. Overall, for two reviews, the agreement was 86.6% (95% CI: 80.6% to 91.8%). It ranged from 75.0% (95% CI: 50.0% to 93.8%) in the Humanities panel to 91.3% (95% CI: 78.3% to 100%) in the Social Sciences panel. Again, there was no evidence for differences in agreement between panels (p=0.51). Both for ERB evaluation with three and two reviewers, the agreement was slightly higher for women than for men (p>0.70, table 3).

In table 4, we calculated agreement separately for the triage categories: Fund (F), Discuss (D) and Reject (R). With the ERB evaluation based on three reviewers, agreements for F and R were close to 100% (97.3% and 96.7%, respectively) but considerably lower for D: 64.2% (95% CI: 52.2% to 76.1%), with p<0.001 for differences in agreement across categories. For ERB evaluation with two reviewers (the two panel members), the agreement was 100% for F and R, but 73.1% (95% CI: 62.7% to 83.6%) for D, with p<0.001 for differences in agreement.

### Random selection in TPM and ERB evaluation

With the standard TPM evaluation, only 8 (11.9%) of the 67 applicants in the D group, or 8 (6.0%) of 134 applicants were entered into a lottery of whom 4 were funded. With the simulated ERB evaluation based on three reviewers, 19 (14.2%) of the 134 applicants would have entered the lottery, and with the ERB with two reviewers 23 (17.2%) applications would have been subjected to random selection.

### Cost and time savings

We determined the resources that could be saved with the use of an ERB evaluation compared with the TPM. By comparison with the current valid TPM evaluation procedure for the Postdoc.Mobility, we calculated that about US$91 000 related to the holding of meetings could have been saved if an ERB evaluation had been used for the two Postdoc.Mobility calls in 2019. This saving corresponds to 55% of total costs. Moreover, the holding of all panel sessions in 2019 amounted to 31 meeting hours, a significant workload that could have been avoided using the ERB approach. Lastly, the funding decisions could have been communicated at least 1 month earlier with ERB,

**Table 2** Agreement between the simulated ERB evaluation and the TPM format, by discipline

| Discipline | N | Funded by TPM | | Agreement (%) (95% CI) |
|---|---|---|---|---|
| **Funded by ERB (three reviewers)*** | | Yes | No | |
| All disciplines | Yes | 59 | 13 | 80.6 |
| | No | 13 | 49 | (73.9 to 87.3) |
| Humanities | Yes | 7 | 2 | 75.0 |
| | No | 2 | 5 | (50.0 to 93.8) |
| Social Sciences | Yes | 9 | 3 | 73.9 |
| | No | 3 | 8 | (56.5 to 91.3) |
| STEM | Yes | 15 | 4 | 77.1 |
| | No | 4 | 12 | (62.9 to 91.4) |
| Biology | Yes | 18 | 3 | 85.0 |
| | No | 3 | 16 | (72.5 to 95) |
| Medicine | Yes | 10 | 1 | 90.0 |
| | No | 1 | 8 | (75 to 100) |
| P value | | | | 0.58 |
| **Funded by ERB (two reviewers)†** | | | | |
| All disciplines | Yes | 63 | 9 | 86.6 |
| | No | 9 | 53 | (80.6 to 91.8) |
| Humanities | Yes | 7 | 2 | 75.0 |
| | No | 2 | 5 | (50.0 to 93.8) |
| Social Sciences | Yes | 11 | 1 | 91.3 |
| | No | 1 | 10 | (78.3 to 100) |
| STEM | Yes | 16 | 3 | 82.9 |
| | No | 3 | 13 | (68.6 to 94.3) |
| Biology | Yes | 19 | 2 | 90.0 |
| | No | 2 | 17 | (80 to 97.5) |
| Medicine | Yes | 10 | 1 | 90.0 |
| | No | 1 | 8 | (75 to 100) |
| P value | | | | 0.51 |

P values for differences in agreement across disciplines from $\chi^2$ test.
*Two of the three expert reviewers were also members of the evaluation panel.
†Both expert reviewers were also members of the evaluation panel.
ERB, expert review based; N, number of applications; STEM, Science, Technology, Engineering, Mathematics; TPM, triage-panel meeting.

reducing the time to notification by about 20% compared with TPM.

## DISCUSSION

In this comparative study of the evaluation of early-career funding applications, we found that the simulated funding outcomes of a simplified, ERB approach agreed well with the official funding outcomes based on the standard, time-tested TPM format. Applications for fellowships covered a wide range of disciplines, from the Humanities and Social Sciences to STEM, Biology and Medicine. The agreement was very high for proposals which, in the TPM evaluation, were either allocated to the fund or reject categories, but lower in the middle category of proposals that were discussed by the panels.

More applicants entered the lottery with the simplified ERB approach than with TPM evaluation. Finally, the simplified ERB evaluation approach was associated with a substantial reduction in costs. Overall, our results support the notion that a sound evaluation of early-career funding applications is possible with an ERB approach.

Although panel review is considered as a 'de facto' standard, the consistency of decisions from panels has been shown to be limited. For example, previous work by Cole et al,[17] Hodgson,[18] Fogelholm et al[11] and Clarke et al[19] found an agreement of 65% to 83% between two independent panels evaluating the same set of applications. Thus, in these studies, the funding outcome also depended on the panel that evaluated the application, and not only on the scientific content. Against this

**Table 3** Agreement between the simulated ERB evaluation and the TPM format, by gender

| Gender | | Funded by TPM | | Agreement (%) (95% CI) |
|---|---|---|---|---|
| Funded by ERB (three reviewers)* | | Yes | No | |
| Women | Yes | 24 | 3 | 83.3 |
| | No | 5 | 16 | (72.9 to 93.8) |
| Men | Yes | 35 | 10 | 79.1 |
| | No | 8 | 33 | (69.8 to 87.2) |
| P value | | | | 0.71 |
| Funded by ERB (two reviewers)† | | | | |
| Women | Yes | 24 | 1 | 87.5 |
| | No | 5 | 18 | (77.1 to 95.8) |
| Men | Yes | 39 | 8 | 86.0 |
| | No | 4 | 35 | (77.9 to 93.0) |
| P value | | | | 0.99 |

P values for differences in agreement across genders from $\chi^2$ test.
*Two of the three expert reviewers were also members of the evaluation panel.
†Both expert reviewers were also members of the evaluation panel.
ERB, expert review based; N, number of applications; STEM, Science, Technology, Engineering, Mathematics; TPM, triage-panel meeting.

background, the agreement of over 80% between ERB and TPM in this study is remarkable. Among the different discipline-specific review panels, our results showed a slightly lower agreement in the Humanities and Social Sciences compared with Life Sciences and Medicine. These differences did not reach conventional levels of statistical significance but were in line with previous findings reported by Pina et al.[13]

In the middle group of applications based on the triage step of TPM, the agreement was lower; 64% with three reviewers and 73% with the two reviewers. This is not surprising considering the results from previous studies

**Table 4** Agreement between the simulated ERB evaluation and the TPM format, by triage results

| Triage result | | Funded by TPM | | Agreement (%) (95% CI) |
|---|---|---|---|---|
| Funded by ERB (three reviewers)* | | Yes | No | |
| Fund (F) | Yes | 36 | 0 | 97.3 |
| | No | 1 | 0 | (91.9 to 100) |
| Discuss (D) | Yes | 23 | 12 | 64.2 |
| | No | 12 | 20 | (52.2 to 76.1) |
| Reject (R) | Yes | 0 | 1 | 96.7 |
| | No | 0 | 29 | (90.0 to 100) |
| P value | | | | <0.001 |
| Funded by ERB (two reviewers)† | | | | |
| Fund (F) | Yes | 37 | 0 | 100 |
| | No | 0 | 0 | |
| Discuss (D) | Yes | 26 | 9 | 73.1 |
| | No | 9 | 23 | (62.7 to 83.6) |
| Reject (R) | Yes | 0 | 0 | 100 |
| | No | 0 | 30 | |
| P value | | | | <0.001 |

P values for differences in agreement across triage groups from $\chi^2$ test.
*Two of the three expert reviewers were also members of the evaluation panel.
†Both expert reviewers were also members of the evaluation panel.
ERB, expert review based; N, number of applications; STEM, Science, Technology, Engineering, Mathematics; TPM, triage-panel meeting.

that suggest that peer review has difficulties in discriminating between applications that are neither clearly excellent nor clearly non-competitive.[20–22] Agreement between ERB and TPM was also generally lower with ERB using three reviewers than with ERB with two reviewers. An additional reviewer may introduce a different viewpoint. Also, the third reviewer was not a member of the corresponding panel, and not involved in previous panel discussions, which have led to some degree of calibration between assessments of panel members. Such calibration is more difficult to achieve with a remote, ERB approach. However, information and briefing sessions could be held to compensate for the lack of FTF panel meetings. Of note, previous studies reported that reviewers appreciated the social aspects and the camaraderie in FTF settings and that physical meetings are important for building trust among the evaluators.[8 9]

We found that the panel discussions in the TPM format resulted in higher success rates for women compared with the ERB format. Gender equality is a key concern at the SNSF, which is committed to promoting women in research. The panels will have been aware of the under-representation of female researchers in certain areas, for example, the STEM disciplines, and the SNSF's agenda to promote women. It is, therefore, possible that during the panel deliberations and for funding decisions, the gender of applicants was taken into account in addition to the quality of the proposal.

We estimated that about US$91 000 could have been saved for the two Postdoc.Mobility calls in 2019 if they had been evaluated by ERB rather than by TPM. The meeting costs represented about 55% of the total evaluation costs. In other words, the ERB evaluation based on the two panel reviewers would have cut the expenses by more than half. The experience described here with the junior Postdoc.Mobility fellowship scheme indicates that substantial cost savings could also result from simplifications in the evaluation of other funding instruments at the SNSF. Also, ERB would have reduced the time to communication of funding decisions, and unsuccessful applicants could thus have planned their next steps earlier. However, any such changes need to be considered carefully. The quality of the evaluation should not be allowed to be compromised because costs may be reduced.

To the best of our knowledge, the Health Research Council of New Zealand (HRC-NZ),[23] the Volkswagen Foundation[24] and recently the Austrian Research Fund FWF[25] are the only funders that have used or examined the use of a random selection element in the evaluation process of funding instruments, with a focus on transformative research or unconventional research ideas. The random selection for decisions on applications close to the funding threshold could avoid bias if evaluation criteria do not allow any further differentiation for a small set of similarly qualified applications.[22 26] The applicants were informed about the possible random selection and the evaluation process thus complied with the San Francisco Declaration on Research Assessment,[27] which states

that funders must be explicit about assessment criteria. In this context, evaluation criteria could be weighted, or additional strategic criteria be used in the selection process if defined a priori and communicated to applicants. However, weighting or additional criteria could also lead to tied applications and thus require a lottery decision. There was some reservation on the random selection approach among some panel members, but acceptance grew over time. Of note, panels applied the random selection only in a few cases, in 8 (6.0%) of 134 applications. In the context of the Explorer Grant scheme of the HRC-NZ, Liu *et al*[28] recently reported that most applicants agreed with the use of a random selection. In this study, the SNSF received a few unsolicited questions about the procedure but otherwise no negative or positive reactions to the use of random selection were received from applicants.

Our study has several limitations. It addressed the specific context of the SNSF Postdoc.Mobility funding scheme and results may not be generalisable to other funding instruments. The sample size was relatively small, and the study lacked statistical power, for example, to examine differences in agreement between TPM and ERB evaluation across disciplines. The two evaluation methods were not independent, since the two assessments of the panel reviewers were used for both methods. We were relying on reviewer evaluation scores which might not always perfectly reflect the quality of the proposed project, might be biased and depend on the reviewers' previous experience with grant evaluation. However, our study design allowed us to investigate the impact of panel meetings on funding outcomes compared with an ERB approach. This study provides further insights into peer review and a modified lottery approach selection in the context of the evaluation of fellowship applications. More research on the limitations inherent in peer review and grant evaluation is urgently needed. Funders should be creative when investigating the merit of different evaluation strategies.[29]

## CONCLUSIONS

In conclusion, we simulated an ERB approach in the evaluation of the junior Postdoc.Mobility funding scheme at the SNSF and compared the funding outcome to the standard TPM format, which has been in use for many years. We found an overall high agreement between the two methods. Discrepancies were mainly observed in the middle group of applications that were discussed in the panel meetings. Based on the evidence that peer review has difficulties in making fine-grained differentiations between meritorious applications,[20–22] we are unsure which method performs better. Our findings indicate that the ERB approach represents a viable evaluation method for the Postdoc.Mobility selection process that could save cost and time which could be invested in science and research.

**Correction notice** This article has been corrected since it was published. The data availability statement in the endnotes has been updated.

**Acknowledgements** We thank the Management of the SNSF Administrative Offices for helpful comments on the design of this study. We also thank the SNSF Postdoc.Mobility staff of the Administrative Offices for their excellent support in implementing the additional reviewers used for the study.

**Contributors** MB, KR and ME conceived and designed the experiments. MB and KR performed the experiments. KR and RH analysed the data. MB, KR, ME and RH contributed reagents/materials/analysis tools. MB wrote the initial draft. KR, ME and RH contributed to writing.

**Funding** This research was supported by institutional resources of the SNSF. ME was supported by a special project grant (grant No. 189498).

**Competing interests** None declared.

**Patient and public involvement** Patients and/or the public were not involved in the design, or conduct, or reporting or dissemination plans of this research.

**Patient consent for publication** Not required.

**Ethics approval** The Ethics Committee of the Canton of Bern confirmed that the study does not fall under the Federal Act on Research involving Human Beings. No reviewer, applicant or application can be identified from this study.

**Provenance and peer review** Not commissioned; externally peer reviewed.

**Data availability statement** Data are available in a public, open access repository. An anonymised data set can be found on Zenodo (https://doi.org/10.5281/zenodo.5546361).

**ORCID iDs**
Marco Bieri http://orcid.org/0000-0002-9831-2146
Katharina Roser http://orcid.org/0000-0001-5253-3333
Rachel Heyard http://orcid.org/0000-0002-7531-4333
Matthias Egger http://orcid.org/0000-0001-7462-5132

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
