## [Reviewer comments · BMJ Open]

ARTICLE DETAILS

TITLE (PROVISIONAL)	Face-to-face panel meetings versus remote evaluation of fellowship applications: simulation study at the Swiss National Science Foundation
AUTHORS	Bieri, Marco; Roser, Katharina; Heyard, Rachel; Egger, Matthias

VERSION 1 – REVIEW

REVIEWER	Grote, Helen Chelsea and Westminster Hospital NHS Foundation Trust
REVIEW RETURNED	16-Jan-2021

GENERAL COMMENTS	I very much enjoyed reading this paper. It is a credit to the authors that careful thought has been given to how the process of grant application reviews could be improved and streamlined, without detriment to the applicant. I was particularly impressed that the effect of gender was considered carefully, in order to support gender equality in research. The study was well structured and analysed. I have a few points the authors may wish to consider to further strengthen their paper. Abstract Line 18: Consider changing to 'junior fellowship proposals for postdoctoral research (Postdoc.Mobility)' – international readers may not have heard what postdoc.mobility is. Introduction Line 59 'Following innovations in career funding...' I am unclear what this means. Please could the authors clarify whether this means that there has been a commitment from the Swiss government for increased research funding, or whether a new model of funding allocation by the SNSF means that there is a greater availability of funding for postdoctoral research. Methods No comma needed after 'Both' in line 19 Line 19: I presume that applications were scored only by the panel relevant to the application.
--

Consider revising this sentence to 'Both fellowship and return grants were evaluated according to the same criteria by the designated panel for the application; Humanities, Social Sciences, STEM (Science, Technology, Engineering and Mathematics), Biology or Medicine.

Figure 1:

Premeeting remote evaluation. You state in methods that 'each proposal was independently reviewed and scored by two panel members'. I assume that there was no collaboration between the panel members at this stage, and that discussion between panel members only occurred after triage in the TPM method.

If this is correct, please amend figure 1 so that the box 'premeeting remote evaluation: 2 panel reviews per application' is changed to 'premeeting remote evaluation: independent review by two panel members'

This helps to clarify there was no collaboration between panel members at this stage in the review process.

Discussion:

There are a few points the reviewers may wish to consider to strengthen the discussion, if the word count allows.

You mention the difficulty of discriminating between applications that are neither clearly excellent nor clearly non competitive. Could weighting of the original evaluation criteria help in a tie-break, rather than a lottery process? Or could a third reviewer be introduced in the ERB process where funding scores between the two experts different by more than a designated number of points, rather than for all reviews?

I note that ERB evaluation with two reviewers the agreement was 100% for F and R, and while a greater proportion of applications 23 (17.2%) would have been entered into the lottery, none of these would have been the 'F' applications. By contrast, the introduction of a third reviewer, meant that agreement for F and R were still high (97.3% and 96.7%) but lower for D. 19 applicants would have been entered into the lottery. Would there be a risk that a potentially 'F' application could end up in the lottery and be rejected?

Benefits to applicants: You mention the benefit of cost savings, and time savings for panel members and SNSF. Are there any benefits to the applicants here? Would it significantly reduce the time they have to wait for outcome of their grant application? (and thereby allow time to pursue other applications/ make other arrangements if unsuccessful). Could the cost savings be reinvested into increasing the number of fellowships/ amount of funding available for research?

REVIEWER	Wallis, Lee A. University of Cape Town, Division of Emergency Medicine
REVIEW RETURNED	22-Jan-2021

GENERAL COMMENTS	Thank you for requesting my review of this article. I regret to say that I recommend rejection of the article. There are a few key reasons for this, including a need for improved description of methodology, limited generalisability of results to BMJ Open's international audience, a small sample size, and weak discussion of findings. Below are a few key recommendations to improve the paper: Overall The title is awkward. I recommend rewording it to: "How to best evaluate applications for junior fellowships? A comparison of remote evaluation and face-to-face panel meetings." In line with BMJ Open style, spell out numbers <10. Abstract Reading the abstract before the full text, it is difficult to understand the purpose, methodology, and results of this work. Authors should rework the abstract and consider having someone external to the research team review it to ensure clarity prior to resubmission. A sentence providing context for the objectives would be useful. International readers are in need of more information regarding the "Postdoc.Mobility" fellowship – a junior fellowship is not universal and needs more context here. Strengths and limitations This section requires additional consideration – at the moment, it does not stand alone and relies quite a bit on readers having already read the abstract and/or full text. It would be of particular use to highlight this study's cost-saving findings and conclusion that it is not clear which method is ultimately better. Introduction The introduction is concise and effectively motivates the study. It would be useful to know a bit more about the Postdoc.Mobility scheme, so that generalisability of results can be better informed. Methods The methods section would benefit from more clarity surrounding the TPM and ERB methods. ERB is only mentioned towards the end, and it is not clear as written how it differs from TPM. Additionally, the data analysis methods are in need of motivation. There are multiple gold standard tests for binary categorical data agreement, including McNemar's and Cohen's kappa. Furthermore, the authors could have looked in more depth at the agreement
--

	of the range of numerical scores with a weighted kappa or similar test. Presently, it is not clear what type of test was used for overall agreement evaluation.
--	---

VERSION 1 – AUTHOR RESPONSE

Reviewer 1

I very much enjoyed reading this paper. It is a credit to the authors that careful thought has been given to how the process of grant application reviews could be improved and streamlined, without detriment to the applicant. I was particularly impressed that the effect of gender was considered carefully, in order to support gender equality in research.

Response: Thank you for the comment.

The study was well structured and analysed. I have a few points the authors may wish to consider to further strengthen their paper.

Abstract, Line 18:

Consider changing to 'junior fellowship proposals for postdoctoral research (Postdoc.Mobility)' – international readers may not have heard what postdoc.mobility is.

Response: Thank you for the suggestion; we agree. We changed the text to "'junior fellowship proposals for postdoctoral research ("Postdoc.Mobility")" as proposed (Abstract, line 18, p. 3).

Introduction, Line 59:

'Following innovations in career funding...' I am unclear what this means. Please could the authors clarify whether this means that there has been a commitment from the Swiss government for increased research funding, or whether a new model of funding allocation by the SNSF means that there is a greater availability of funding for postdoctoral research.

Response: Thank you. We changed "Following innovations in career funding,..." to "Following changes to the career-funding portfolio,..." (Introduction, line 59, p. 5).

Methods

No comma needed after 'Both' in line 19

Response: We removed the comma in line 19, p. 6.

Line 19:

I presume that applications were scored only by the panel relevant to the application. Consider revising this sentence to 'Both fellowship and return grants were evaluated according to the same criteria by the designated panel for the application; Humanities, Social Sciences, STEM (Science, Technology, Engineering and Mathematics), Biology or Medicine.

Response: Many thanks, we implemented your suggestion as follows: "Both fellowship and return grants were evaluated according to the same criteria by one of five panels: Humanities, Social Sciences, STEM (Science, Technology, Engineering and Mathematics), Biology or Medicine" (Methods, line 19, p. 6).

Figure 1:

Premeeting remote evaluation. You state in methods that 'each proposal was independently reviewed and scored by two panel members'. I assume that there was no collaboration between the panel members at this stage, and that discussion between panel members only occurred after triage in the TPM method.

If this is correct, please amend figure 1 so that the box 'premeeting remote evaluation: 2 panel reviews per application' is changed to 'premeeting remote evaluation: independent review by two panel members'

This helps to clarify there was no collaboration between panel members at this stage in the review process.

Response: Yes, this is correct; thank you for the suggestion. We adapted the figure accordingly.

Discussion:

There are a few points the reviewers may wish to consider to strengthen the discussion, if the word count allows.

You mention the difficulty of discriminating between applications that are neither clearly excellent nor clearly non competitive. Could weighting of the original evaluation criteria help in a tie-break, rather than a lottery process? Or could a third reviewer be introduced in the ERB process where funding scores between the two experts differ by more than a designated number of points, rather than for all reviews?

Response: Weighting of original evaluation criteria (or even secondary criteria, such as, e.g., gender) could be used to break ties, but only when transparently disclosed in the evaluation process beforehand (DORA principle regarding transparency). However, also the use of weighting/secondary criteria could lead to a set of tied applications and require a lottery decision. Yes, a third reviewer could be introduced for the ERB where funding scores between the two experts differ, but also this does not necessarily exclude a tied situation. We added the following statement (Discussion, line 52, p. 11): "In this context, evaluation criteria could be weighted, or additional strategic criteria be used in the selection process if defined a priori and communicated to applicants. However, weighting or additional criteria could also lead to tied applications and thus require a lottery decision."

I note that ERB evaluation with two reviewers the agreement was 100% for F and R, and while a greater proportion of applications 23 (17.2%) would have been entered into the lottery, none of these would have been the 'F' applications. By contrast, the introduction of a third reviewer, meant that agreement for F and R were still high (97.3% and 96.7%) but lower for D. 19 applicants would have been entered into the lottery. Would there be a risk that a potentially 'F' application could end up in the lottery and be rejected?

Response: A third reviewer adds an additional viewpoint and it is possible that an "F" application (based on two reviews) is shifted downward into the lottery group because of a negative third review. However, the opposite scenario is also possible; an "R" application (based on two reviews) is shifted upward into the lottery group. Once entered into the lottery, the outcome obviously depends on chance. In our study over the total sample of 134 applications, only one single "F" application moved down and entered into the lottery due to a third negative review (and was subsequently rejected by drawing lots), and only one single "R" application moved up into the lottery (and was subsequently rejected by drawing lots). No revision was made in the manuscript.

Benefits to applicants: You mention the benefit of cost savings, and time savings for panel members and SNSF. Are there any benefits to the applicants here? Would it significantly reduce the time they have to wait for outcome of their grant application? (and thereby allow time to pursue other applications/ make other arrangements if unsuccessful). Could the cost savings be reinvested into increasing the number of fellowships/ amount of funding available for research?

Response: Thank you for this input. With ERB evaluation, no panel meetings must be prepared and held. We indeed anticipate that the duration until the communication of the decisions to the applicants could be reduced from five (valid evaluation format TPM) to four months (ERB) by at least a month. Another benefit for the applicant is a more transparent and fairer evaluation procedure. The saved costs of about USD 91,000 would roughly correspond to one fellowship award, if existing budgets of funding and evaluation costs could be balanced in a more flexible manner (which is not yet the case at SNSF).

We changed the subsection heading from "Cost savings" to "Cost and time savings" (Results, line 26, p. 9). In the same subsection, on line 38, we added the sentence "Lastly, the funding decisions could have been communicated at least one month earlier with ERB, reducing the time to notification by about 20% compared to TPM." In the Discussion section, line 32, p. 11, we added the sentence "Also, ERB would have reduced the time to communication of funding decisions, and unsuccessful applicants could thus have planned their next steps earlier".

Reviewer 2

I regret to say that I recommend rejection of the article. There are a few key reasons for this, including a need for improved description of methodology, limited generalisability of results to BMJ Open's international audience, a small sample size, and weak discussion of findings. Below are a few key recommendations to improve the paper:

Response: Our study makes an important contribution that supports the use of more cost-effective evaluation procedures and we are convinced that this will be of interest to the international readership of BMJ Open. We acknowledge that our study lacks statistical power, but we strongly disagree that this invalidates the study as a whole. The sample of 134 fellowship proposals was small, but it still allowed for identifying meaningful differences. BMJ Open published a study on peer review relying on an even smaller sample size of 72 grant applications (Herbert et al., doi:10.1136/bmjopen-2015-008380) and in PNAS a study on the NIH review process relying on 25 grant applications was published (Pier et al., doi:10.1073/pnas.1714379115). The "limited generalizability of results" is not a valid criticism in our view. Different funding schemes with varying

procedures of selection obviously exist across funders. Thus, any study on peer review inevitably refers to a specific setting, which cannot be considered as a flaw of the study itself. On the contrary, more studies on peer review are needed to draw more generalisable conclusions. The reviewer further mentions "a weak discussion of results", but no details are provided, which is not helpful.

Overall

The title is awkward. I recommend rewording it to: "How to best evaluate applications for junior fellowships? A comparison of remote evaluation and face-to-face panel meetings."

In line with BMJ Open style, spell out numbers <10.

Response: We agree that our initial title did not fully comply with BMJ Open's guidelines. We changed the title to "Face-to-face panel meetings versus remote evaluation of fellowship applications: simulation study at the Swiss National Science Foundation".

Abstract

Reading the abstract before the full text, it is difficult to understand the purpose, methodology, and results of this work. Authors should rework the abstract and consider having someone external to the research team review it to ensure clarity prior to resubmission.

A sentence providing context for the objectives would be useful. International readers are in need of more information regarding the "Postdoc.Mobility" fellowship – a junior fellowship is not universal and needs more context here.

Response: We provide a concise and structured abstract according to BMJ Open's submission guidelines. Purpose, methodology and results are described in dedicated sections. In response to the reviewer's suggestion to provide a sentence on the context for the objectives, we changed the first part of the starting sentence from "To test a simplified evaluation of fellowship proposals..." to "To compare a cost-saving, simplified evaluation of fellowship proposals with the official evaluation by analyzing..." (Abstract, line 9, p. 3).

The reviewer also states that more information regarding "Postdoc.Mobility" is needed for international readers. We agree. We refined to "junior fellowship proposals for postdoctoral research ("Postdoc.Mobility")" as suggested by reviewer 1 (Abstract, line 18, p. 3). Since space is very limited in the abstract section, we give some further information in the introduction section (as also suggested by reviewer 2 below).

Strengths and limitations

This section requires additional consideration – at the moment, it does not stand alone and relies quite a bit on readers having already read the abstract and/or full text. It would be of particular use to highlight this study's cost-saving findings and conclusion that it is not clear which method is ultimately better.

Response: The reviewer seems to misinterpret the purpose of the "strengths and limitations" section. BMJ Open's submission guidelines state regarding "strengths and limitations": "An Article Summary, placed after the abstract, consisting of the heading 'Strengths and limitations of this study', and containing up to five short bullet points, no longer than one sentence each, that relate specifically to the methods. They should not include the results of the study." Thus, the strengths and limitations section follows the

abstract and does not stand alone. The reviewer suggests "to highlight this study's cost-saving findings and conclusion that it is not clear which method is ultimately better." However, these results do not refer to the "strengths and limitations" section. No changes were made to the manuscript.

Introduction

The introduction is concise and effectively motivates the study. It would be useful to know a bit more about the Postdoc.Mobility scheme, so that generalisability of results can be better informed.

Response: We agree and we added the sentence "The scheme enables junior postdocs to deepen their scientific knowledge and increase their scientific independence during a research stay abroad." (Introduction, line 4, p. 6) to outline the main aim of the fellowship scheme.

Methods

The methods section would benefit from more clarity surrounding the TPM and ERB methods. ERB is only mentioned towards the end, and it is not clear as written how it differs from TPM.

Response: This criticism is vague, the reviewer does not state why more clarity is needed. It is true that ERB is explained at the end because we first describe TPM reference evaluation method. The reviewer mentions that it is not clear how ERB differs from TPM. However, we clearly state that "the simulated alternative procedure [ERB] consisted only of the first step, i.e., was entirely based on the ranking of proposals based on the expert reviews", which is overall rather simple and straightforward. With the aid of Figure 1, we believe that TPM and ERB are presented in a clear and concise way. No revision was made in the manuscript.

Additionally, the data analysis methods are in need of motivation. There are multiple gold standard tests for binary categorical data agreement, including McNemar's and Cohen's kappa. Furthermore, the authors could have looked in more depth at the agreement of the range of numerical scores with a weighted kappa or similar test. Presently, it is not clear what type of test was used for overall agreement evaluation.

Response: Thank you for pointing us to weaknesses in the data analysis section. Given the relatively small data set, we wanted to keep the statistical methods as basic and simple as possible. This is the main reason why we only calculated simple agreement. We now added bootstrap confidence intervals (CI) as it has already been done in similar data analyses (see reference [5] in manuscript). These CIs should give enough information regarding overall agreement. Then, the tests that were additionally performed are Chi-squared tests, which should conclude whether there is evidence for differences in agreement between mutually exclusive groups, notably male vs. female researchers, or the different panels. As these groups are mutually exclusive, e.g. independent, we believe chi-squared tests are fine. The suggested McNemar test is for overall agreement; leading to a comparison of binary outcomes, which is not useful to answer the latter question. We judge the provided CIs as enough information on the "significance" of agreement.

We made adaptations in the Methods section (subsection Data analysis): "We calculated the simple agreement with 95% confidence intervals (CI), which were generated using a

bootstrap algorithm [5]" (line 17, p. 7). We further added "We used chi-squared tests for categorical data to test whether the agreement differed between these mutually exclusive groups." (line 22, p. 7).

In the Abstract section (subsection "Results") and in the Results section (subsection "Agreement between evaluation by ERB or TPM"), we adapted the CI-values based on the bootstrap algorithm. Eventually, CI-values were adapted in Tables 2-4.

Reviewer 3 (unsolicited review by Prof. Adrian Barnett via bioRxiv)

This is a useful experiment given the shortage of experiments into funding. As Guthrie et al (reference #1) stated: "We need to overcome the reluctance of funders and scientists to acknowledge the uncertainties intrinsic to allocating research funding, and encourage them to experiment with peer review and other allocation processes". The results are broadly supportive of a simpler and cheaper peer review system.

The agreement between reviewers was not adjusted for chance (e.g, using Gwet's statistic). I agree with this approach as the raw agreement is what researchers are interested in (their only question is always, "Was I funded or not?"). We can account for chance by setting a threshold for an acceptable difference, e.g., an agreement of 75%. This threshold would ideally be based on discussions with the research community.

The differences in agreement were tested using chi-squared, but these are paired categorical data and so I think McNemar's test would be better. Although I'm not sure that p-values are useful given the sample size and the potential for a p-value of 0.05 to be interpreted as demonstrating equivalence. I would focus on the confidence intervals and whether they rule out an important difference in agreement.

Response: We share your cautiousness towards p-values. McNemar's test would relate to testing overall agreement (\diamond leading to a comparison of binary outcomes). As you suggest, we already provide CI, which give enough insight on the importance of agreement. However, we would like to use a test to see whether the overall agreement differs between mutually exclusive groups (gender, panels, ...). Therefore, we believe chi-squared test should be fine, as these groups are independent.

The authors use Wald intervals but the sample size is small and the proportion is sometimes close to one, hence the normal assumption may start to be strained. I would consider using a bootstrap interval.

Response: Thank you for the suggestion; we changed the Wald to Bootstrap intervals.

Although face-to-face meetings for peer reviewers may increase trust they also are a networking opportunity and could disadvantage those not invited or unable to attend (e.g., researchers caring for children). It is also a great learning opportunity for the reviewers about what makes a good application.

Minor comments

- Table 1 shows summary statistics not "the distribution"

Response: Thank you for the comment; we changed "Table 1 shows the distribution of applications..." to "Table 1 shows the summary statistics of applications..." (Results, line

16, p. 8).

- "no negative or positive reactions to the use of random selection were received from applicants" but was feedback asked for or were there only unsolicited comments?

Response: Thank you for the comment. No, the SNSF did not formally ask for feedback but received some unsolicited questions regarding the random selection procedure. We have clarified this as follows: "In this study, the SNSF received a few unsolicited questions about the procedure but otherwise no negative or positive reactions to the use of random selection were received from applicants." (Discussion, line 59, p. 11).

- The success rates here are very high success rate compared with other schemes. This may put less pressure on the system and allow it to conduct more novel experiments such as modified lotteries.

We agree with this comment but did not make any changes to the manuscript.

Lastly, we made some minor adaptations in the "Acknowledgments" and "Funding" sections.

Thank you very much for considering our work for publication in BMJ Open.

VERSION 2 – REVIEW

REVIEWER	Grote, Helen Chelsea and Westminster Hospital NHS Foundation Trust
REVIEW RETURNED	07-Apr-2021

GENERAL COMMENTS	Reviewer 1 comments: Thank you for addressing my comments. The revised paper is much improved and I would recommend publication. I have only two comments: 1. Consider rewording 'objectives'. At present it reads like a translation into English and does not link well with the title. It is not clear to the reader from this what the 'official evaluation' is, and does not highlight that the remote evaluation proposal is a new one. A potential form of wording could be similar to that suggested below: To trial a simplified, time and cost-saving method for remote evaluation of fellowship applications and compare this with existing panel review processes by analyzing concordance between funding decisions, and the use of a lottery-based decision method for proposals of similar quality.2. I could not see a copy of Figure 1, but presume this has been appropriately amended as per the author response (I suggested the box 'premeeting remote evaluation: 2 panel reviews per application' should be changed to 'premeeting remote evaluation: independent review by two panel members' to clarify that there was no collaboration between panel members at this stage in the review process.
---

REVIEWER	Wallis, Lee A. University of Cape Town, Division of Emergency Medicine
REVIEW RETURNED	04-Apr-2021

GENERAL COMMENTS	Thank you. the paper reads better, and it will be interesting to see in print.
--

VERSION 2 – AUTHOR RESPONSE

Reviewer 1

Thank you for addressing my comments. The revised paper is much improved and I would recommend publication.

I have only two comments:

1. Consider rewording 'objectives'.

At present it reads like a translation into English and does not link well with the title. It is not clear to the reader from this what the 'official evaluation' is, and does not highlight that the remote evaluation proposal is a new one. A potential form of wording could be similar to that suggested below:

To trial a simplified, time and cost-saving method for remote evaluation of fellowship applications and compare this with existing panel review processes by analyzing concordance between funding decisions, and the use of a lottery-based decision method for proposals of similar quality.

Response: Thank you for the comment. We agree and replaced the former statement "To compare a cost-saving, simplified evaluation of fellowship proposals with the official evaluation by analyzing the agreement of funding decisions, and to examine the use of a lottery-based decision for proposals of similar quality." with your suggestion "To trial a simplified, time and cost-saving method for remote evaluation of fellowship applications and compare this with existing panel review processes by analyzing concordance between funding decisions, and the use of a lottery-based decision method for proposals of similar quality."

2. I could not see a copy of Figure 1, but presume this has been appropriately amended as per the author response (I suggested the box 'premeeting remote evaluation: 2 panel reviews per application' should be changed to 'premeeting remote evaluation: independent review by two panel members' to clarify that there was no collaboration between panel members at this stage in the review process.

Response: Indeed, we amended the figure according to your suggestion and uploaded it in the system. We do not know why it was not visible for you.

Reviewer 2

Thank you. The paper reads better, and it will be interesting to see in print.

Response: Many thanks for the comment.